# Simulation and Experimental Study on Residual Stress Distribution in Titanium Alloy Treated by Laser Shock Peening with Flat-Top and Gaussian Laser Beams

**DOI:** 10.3390/ma12081343

**Published:** 2019-04-24

**Authors:** Xiang Li, Weifeng He, Sihai Luo, Xiangfan Nie, Le Tian, Xiaotai Feng, Rongkai Li

**Affiliations:** 1Science and Technology on Plasma Dynamics Laboratory, Air Force Engineering University, Xi’an 710038, China; 651486396@163.com (X.L.); hehe_coco@163.com (W.H.); dreamcatcher_le@163.com (L.T.); shmily9551@163.com (X.F.); 3140103347@zju.edu.cn (R.L.); 2Institute of Aeronautics Engine, School of Mechanical Engineering, Xi’an Jiaotong University, Xi’an 710049, China; 3School of Mechanical and Power Engineering, East China University of Science and Technology, Shanghai 200237, China; skingkgd@163.com

**Keywords:** laser shock peening, spatial energy distribution, flat-top laser beam, Gaussian laser beam, finite element mode, residual stress

## Abstract

The residual stress introduced by laser shock peening (LSP) is one of the most important factors in improving metallic fatigue life. The shock wave pressure has considerable influence on residual stress distribution, which is affected by the distribution of laser energy. In this work, a titanium alloy is treated by LSP with flat-top and Gaussian laser beams, and the effects of spatial energy distribution on residual stress are investigated. Firstly, a 3D finite element model (FEM) is developed to predict residual stress with different spatial energy distribution, and the predicted residual stress is validated by experimental data. Secondly, three kinds of pulse energies, 3 J, 4 J and 5 J, are chosen to study the difference of residual stress introduced by flat-top and Gaussian laser beams. Lastly, the effect mechanism of spatial energy distribution on residual stress is revealed.

## 1. Introduction

In most case, fatigue cracks are likely to initiate in the materials surface. Therefore, optimization of the materials surface integrity can effectively improve the reliability of parts and prolong the service life of components [1]. Residual stress is one main factor of surface integrity that affects the fatigue performance of metallic alloys. Surface modification techniques are effective methods when introducing the compressive residual stress in material surfaces, such as shot peening [2], surface mechanical attrition treatment [3] and cold rolling [4]. Laser shock peening (LSP) is an advanced modification technique used to introduce high amplitude compressive residual stress and improve fatigue performance [5,6,7]. Compared with shot peening, the depth of compressive residual stress induced by LSP can extend to 1 mm, so the processed materials have a robust resistance to crack initiation and propagation, and the fatigue performance is markedly improved [8,9].

Up to now, many research studies have been attempted to reveal the formation mechanism and characteristic of residual stress induced by LSP through experiment and finite element model (FEM) methods. Sun et al. [10] investigated the effects of different pulse energies (3, 5 and 7 J) on the surface morphology and residual stress field of TC17 titanium alloy by FEM and experiment methods. The results showed that the higher the energy was, the greater the compressive residual stress induced. Nie et al. [11] studied the effects of residual stress distribution on TC6 titanium alloy with multiple LSP treatment, and the results showed that the compressive residual stress increased with the increment of the LSP impact. Luo at al. [12] used four kinds of overlapping rates (30%, 50%, 70% and 90%) to investigate the uniformity and the affected depth of residual stress, and the results showed that deeper and more uniform plastic deformation was generated under larger overlapping rates.

Significant work has been conducted in residual stress distribution induced by different LSP parameters such as power density [13], LSP impact [14] and overlapping rate [15]. These investigations mainly focused on the traditional physical LSP parameters. In fact, the spatial energy distribution can also result in different residual stress distributions after LSP, and the Gaussian and flat-top laser were chosen as the energy source during LSP from the open literature [16,17,18]. The energy distribution of Gaussian spatial energy laser beam (GSELB) in space has a certain gradient, and the energy distribution of flat-top spatial energy laser beam (FSELB) over the radial direction of the spot is uniform. Hence, further investigation of the effects of residual stress induced by LSP with different spatial energy distributions is still necessary.

In this work, a three-dimensional finite element model (FEM) using ABAQUS software is developed to predict the residual stress generated by GSELB and FSELB. Subsequently, the simulated residual stress is analyzed and validated. Three kinds of pulse energies, 3 J, 4 J and 5 J, are chosen to investigate the residual stress distribution induced by GSELB and FSELB. Finally, the effect mechanism of spatial energy distribution on residual stress is revealed.

## 2. Materials and Experiments

### 2.1. Material and Components

TC4 titanium alloy in the present study is a kind of α + β type two-phase mid-strength alloy, which has good comprehensive performance and is widely used in aviation material parts. The chemical composition of TC4 titanium alloy is given in Table 1. The mechanical properties of this alloy are given in Table 2 [19].

### 2.2. LSP Experiment

In the LSP process, a water layer with a thickness of about 1 mm was used as the transparent confining layer and an Al foil with a thickness of 100 μm was used as the absorbing layer. The detailed principle of LSP is described by Ye et al. [5]. The sample dimension used in LSP was 30 mm × 30 mm × 6 mm and the LSP processed area was a square region with 15 mm × 15 mm dimensions on one side. The path of LSP and the overlap of spots are shown in Figure 1. The detailed LSP parameters are listed in Table 3.

The LSP experiment was carried out using YD60-M165 and YS80-M165 equipment (TYRIDA, Xi’an, China), respectively. The spatial energy of YD60-M165 was set as Gaussian and that of the YS80-M165 was set as a flat-topped distribution. The corresponding spatial energy distributions and pressure distributions of the laser beams are shown in Figure 2.

### 2.3. Residual Stress Measurement

According to Bragg’s diffraction law, a Proto-LXRD instrument was used to measure the residual stress after LSP using an X-ray diffractometer with the sin^2^*Ψ*-method. Alignment of equipment was checked before each set of measurement using a standard sample (pure Ti in this case) in accordance with ASTM E915-10 [20]. The spot of the X-ray beam was 2 mm. Diffracted Cu–Kα characteristic X-ray from a Ti {213} plane was detected with a 2*θ* of 139°–142°. The X-ray diffraction elastic constants were measured in accordance with ASTM E1426-2014 [21]. Samples were removed layer by layer via electro-polishing with POLISHER 8818 V-3 equipment at room temperature. The polishing solution was composed of 10 vol% perchloric acid (10 vol% HClO_4_) and 90 vol% methanol (90 vol% CH_3_OH). Stress relaxation and stress gradients due to layer removal from electro-polishing corrections were applied to the residual stress values using established procedures in the software with the SAE HS-784 standard [22]. The residual stress on the surface and at greater depths was tested using the X-ray diffraction method. The points were measured along the X direction and the distance between the two points was 2.5 mm. The residual stress at certain depths was tested layer by layer and the average residual stress value of three points at the same horizontal line was taken as the final result.

## 3. FEM Simulation and Procedure

### 3.1. Three-Dimensional FEM Model

A three-dimensional finite element method (FEM) was developed to calculate the residual stress on the target material. Two successive steps, dynamic analysis and static analysis, were used by the ABAQUS/Explicit code and the ABAQUS/Standard code, respectively. ABAQUS/Explicit was first utilized to conduct dynamic calculation, which was implemented to capture the material response. After the dynamic calculation completion, the obtained results were input into ABAQUS/Standard to perform static analysis, and then to output the final result. The FEM was composed of 345,600 C3D8R-type elements with blocks of 0.25 × 0.25 × 0.25 mm, and the dimensions of the target material were the same as those used in the experiment.

It should be noted that the dynamic behavior of TC4 titanium alloy plays an important role in residual stress. In the LSP process, the typical strain is as high as 10^−7^ s^−1^ [23], so the constitutive relation obtained from quasi-static analysis is no longer suitable. The Johnson–Cook (J–C) model considers the work hardening and strain rate hardening effect, which has been widely adopted to simulate the LSP process [17,24,25]. In this work, LSP is assumed to be a purely mechanical process, so the increase in temperature in the substrate material is negligible. This assumption is also presented in various pieces of literature [26,27]. Thus, the temperature softening effect in the J–C model was not considered. The model can be expressed as follows [28]:(1)$$\sigma =(A+B\varepsilon^{n})[1+C\ln(\dot{\varepsilon }/\dot{\varepsilon_{0}})]$$
where *σ* is stress, *ε* is strain and $\dot{\varepsilon }$ is the strain rate. $\dot{\varepsilon_{0}}$ is the reference strain rate. *A*, *B*, *n*, *C* are the model parameters. The J–C constitutive model parameters of TC4 titanium alloy materials are shown in Table 4.

### 3.2. The Distribution of the Shock Wave Pressure

#### 3.2.1. The Time Distribution of the Shock Wave Pressure

The residual stress generated by LSP is dependent on the pressure of the shock wave. The GSELB and FSELB both have a Gaussian distribution in time and the loading spatial pressures of GSELB and FSELB are shown in Figure 1. In order to simplify the calculation, the pressure loading time distribution of the shock wave during the numerical simulation can be simplified to the triangular waveform *P*(*t*), as shown in Figure 3.

Due to the short duration of shock waves (only about a few nanoseconds), the action time of the shock wave on the material is reflected in the impulse amount in the process of LSP, and the material properties are changed. Peyre et al. [30] believed that the action time of the shock wave was about 2 to 3 times of the laser pulse width (20 ns in this paper for the two lasers). In this paper, the shock wave loading time was set as 68 ns and the rising time was set as 20 ns. The pressure loading was observed to rise firstly and then decay, which is conducive to the formation of a steep shock front.

#### 3.2.2. The Spatial and Spatiotemporal Distribution of the Shock Wave Pressure

According to the characteristics of FSELB, it was assumed that the spatial energy is uniform. The pressure can be expressed as:(2)$$P_{1}(r)=P_{center\_1}$$
where *P*_1_(*r*) is the spatial pressure distribution of FSELB, *P_center_*__1_ represents the peak pressure of FSELB. The ideal profile of FSELB is shown in Figure 4a. Figure 4b shows the ideal pressure distribution curve of the GSELB. The spatial pressure distribution of GSELB can be described as shown in Equation (3) [31]:(3)$$P_{2}(r)=P_{center\_2}\exp(\frac{-r^{2}}{2R^{2}})$$
where *r* is the distance from the laser center, *R* is the diameter of GSELB, *P*_2_(*r*) is the function of spatial pressure change with time and *P_center_*__2_ is the peak pressure of GSELB.

The pressure spatiotemporal distribution *P*(*r*,*t*) is composed of *P*(*t*) and *P*(*r*). From what has been discussed above, the pressure spatiotemporal distribution can be described as:(4)$$P_{1}(r,t)=P_{center\_1}\cdot P(t)$$
(5)$$P_{2}(r,t)=P_{center\_2}P(t)\cdot \exp(-\frac{r^{2}}{2R^{2}})$$
where *t* is the time. *P*(*t*) is the time distribution of the shock wave pressure.

A simple pressure model with FSELB under the constraint condition was proposed by Fabbro [32]. The plasma is considered as an ideal gas in the model and the peak pressure of shock wave is estimated as:(6)$$P(t)=0.01\sqrt{\frac{\alpha }{2\alpha +3}}\sqrt{Z(g\cdot cm^{-2}s^{-1})}\sqrt{I(GW/cm^{2})}$$
where *I* is the power density of the laser beam, *P* is the pressure of the laser-induced shock wave, *α* is the efficiency coefficient and *Z* is the reduced shock impedance related to the target and confining medium. *Z* is defined as 2/*Z* = 1/*Z* water + 1/*Z* tape, where *Z* water and *Z* tape represent the acoustic impedance of water and the black tape, respectively.

In this paper, the confining deposit was water and the absorbing layer was black tape. *α* was 0.15 and *Z* was 0.926 × 106 g∙cm^−2^∙s^−1^. The peak pressure of GSELB was calculated according to the literature [30]. In the simulation, the laser wavelength was 1064 nm, the pulse energy was 4 J, the pulse duration was 20 ns and the spot diameter was 2.2 mm.

## 4. Results and Discussion

### 4.1. Residual Stress Distribution of Simulation Results

The residual stress distributions in a single point after LSP with two different kinds of laser beams are shown in Figure 5. The FEM results show that the compressive residual stress induced by LSP with GSELB reached its maximum in the center of the spot and decreased to the periphery. The surface residual stress induced by LSP with FSELB was uniform and the tensile stress appeared at the edge. This is basically in accordance with the actual laser energy distribution shown in Figure 2.

The residual stress distribution of the FEM results is shown in Figure 6. The results show that a considerable compressive residual stress was generated at the surface and at greater depths. A quietly uniform compressive residual stress generated by LSP with GSELB after overlap is shown in Figure 6a, while Figure 6b indicates that the residual stress induced by LSP with FSELB is uneven. In addition, it can be seen that the largest compressive residual stress value produced by GSELB was larger than that of FSELB. This is because the largest deformation was produced by GSELB, and the deformation degree in the center of the treated region was more homogeneous than that treated by FSELB.

A schematic diagram of the plastic deformation after LSP with two different spatial energy distributions is shown in Figure 7. The essence of overlap is the increase of the number of LSP impacts, which results in different deformation degrees in a circular spot. Moreover, the shock times in the LSP-treated areas are different, including one, two, three and four times, as seen in Figure 7, wherein the corresponding areas are presented by different colors. The deformed regions 1 and 2 represent the areas treated by fewer shock times (1–2 times) and more shock times (3–4 times) with GSELB, respectively. The deformed regions 3 and 4 represent the areas treated by fewer shock times (1–2 times) and more shock times (3–4 times) with FSELB, respectively.

Compared to the plastic deformation in the top surface position 1, the plastic deformation in the surface position 2 remained in a circular spot. Therefore, according to the characteristics of the laser beam (Figure 2), the plastic deformation of regions 2 and region 4 accounts for the majority in a single spot. The plastic deformation in region 2 was larger and more uniform than that in region 4 (H3 > H4), and the plastic deformation in region 1 was greater than that in region 3 (H1 > H2). This is why the compressive residual stress induced by LSP with GSELB is higher and more uniform than that with FSELB.

### 4.2. Verification of Residual Stress of Simulation Results

In order to verify the results of the above FEM model, the LSP experiments were carried out with the same parameters as the FEM simulation. The surface residual stress and the residual stress at greater depths are given in Figure 8. The surface residual stress of the FEM was obtained along path 1 (in Figure 6). Seven points were tested by experiments and the distance between each two points was 2.5 mm. At greater depths, the average value of residual stress along the three black arrows (in Figure 6) was obtained as the final simulation value and the experimental tests were consistent with the simulation. Figure 8a shows that surface compressive residual stress generated by LSP with GSELB fluctuated to some extent by using FEM simulation and the results are basically consistent with the experimental data. The difference is that the FEM model simulated a general trend of the residual stress lower than that obtained by the experiments, though similar results were obtained. This is plausibly caused by the overestimation of the strain hardening effect in such a high plastic deformation region [33]. Figure 8b shows that the residual stress distribution at various depths agrees well with experiment. Similar to the results of GSELB, the surface residual stress obtained by FEM exhibited a certain fluctuation, and the experimental data also showed a certain fluctuation, which is consistent with the phenomenon of surface residual stress shown in Figure 8c. This is mainly due to the characteristics of the laser beams and the overlap of spots. The residual stress distribution at various depths is shown in Figure 8d, and the FEM results are consistent with the experiment data. In addition, a larger surface residual stress for GSELB was formed (Figure 8b,d), which was attributed to the larger shock wave pressure under the same laser parameters induced by LSP with GSELB.

### 4.3. Residual Stress Distributions with Different Pulse Energies

To further investigate the effect of residual stress distribution induced by LSP with two laser beams, three different pulse energies (3 J, 4 J and 5 J) were chosen during LSP. The residual stress distribution is shown in Figure 9. When the pulse energy was 3 J, the average value of surface compressive residual stress induced by LSP with GSELB was 656 MPa. The compressive stress values were 777 MPa and 848 MPa for 4 J and 5 J pulse energies, respectively. When the distribution of the laser beam was flat-top, the average values of surface compressive residual stress were 565 MPa, 658 MPa and 732 MPa, respectively. When the energy was the same, the surface compressive residual stress generated by LSP with GSELB was greater than that induced by FSELB. Moreover, the higher the energy, the greater the compressive stress. With the increase of the pulse energy, the surface compressive residual stress produced by both laser beams was increased.

In the cross-section, the value of compressive residual stress reduced gradually with the increase of depth. The depth of compressive residual stress induced by GSELB with 3 J was 0.565 mm. The depths of compressive residual stress were 1.063 mm and 1.305 mm for 4 J and 5 J pulse energies, respectively. The depths of compressive residual stress induced by LSP with FSELB were 0.405 mm, 0.825 mm and 0.976 mm, respectively. These results indicate that compressive residual stress produced by the two kinds of laser beams with the same pulse energy is different. The depth of compressive residual stress induced LSP with GSELB was greater than that with FSELB. With the increase of pulse energy, the compressive residual stress layer both become deeper and deeper.

The residual stress distribution induced by LSP with GSELB and FSELB was significantly different under the same LSP parameters. The compressive residual stress induced by LSP with GSELB was larger than that with FSELB at the surface and at greater depths. This is due to the difference in the peak pressure of the laser-induced shock waves. The peak pressure of the GSELB was 1.5 times that of FSELB under the same parameters [12]. The plastic deformation of the material was affected by the laser-induced shock wave pressure, and the residual stress of the TC4 titanium alloy was determined by the plastic deformation after LSP.

## 5. Conclusions

(1)FEM results show that the residual stress induced by LSP with GSELB and FSELB is different. This is due to the difference in spatial energy distribution and the peak pressure of the shock wave. The predicted residual stress induced by GSELB is considerably larger than that induced by FSELB.(2)The FEM method was used to predict the residual stress distribution with two kinds of laser beams, and good agreements were obtained between the predicted results and the experimental data.(3)Compared with FSELB, the GSELB treatment of TC4 titanium alloy can generate high-level and more uniform compressive residual stress under the same LSP parameters. With the increase of pulse energy, the compressive residual stress is increased both at the surface and at greater depths.

## Reference

## Figures and Tables

**Figure 1 materials-12-01343-f001:**
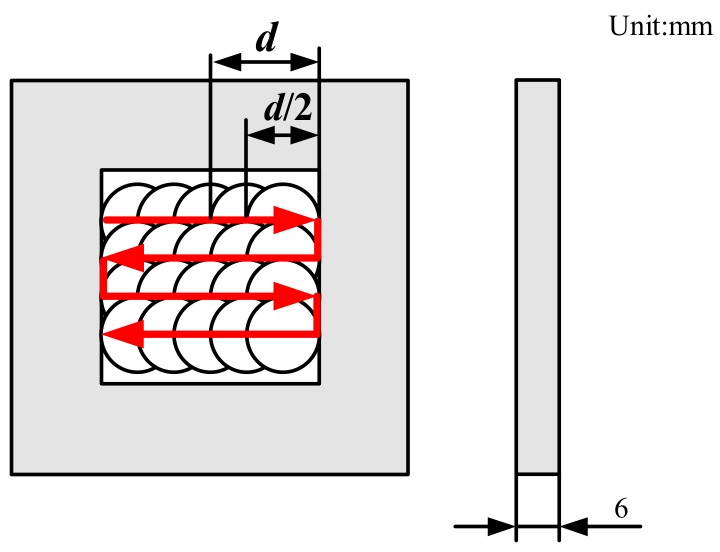
Schematic of the spots overlap and laser shock peening (LSP) path.

**Figure 2 materials-12-01343-f002:**
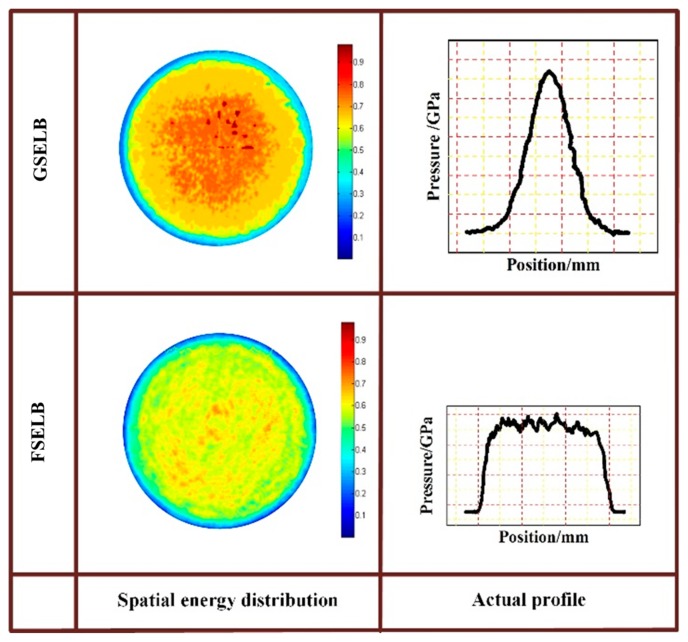
The energy distribution of the laser beams and section profile of pressure.

**Figure 3 materials-12-01343-f003:**
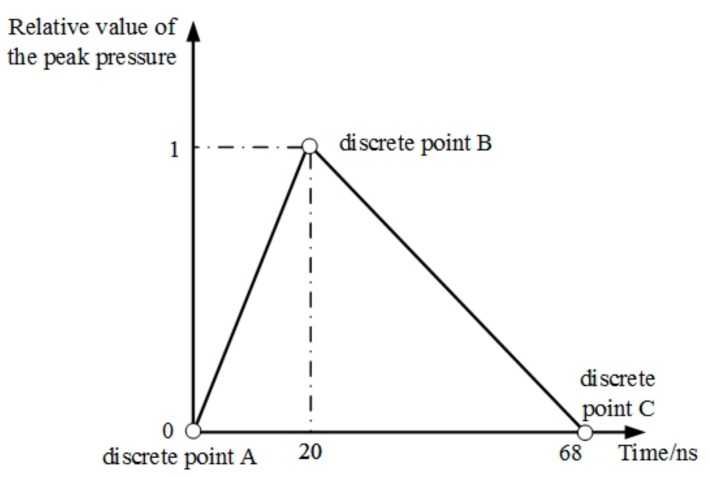
The curve of shock wave pressure loading with pulse width.

**Figure 4 materials-12-01343-f004:**
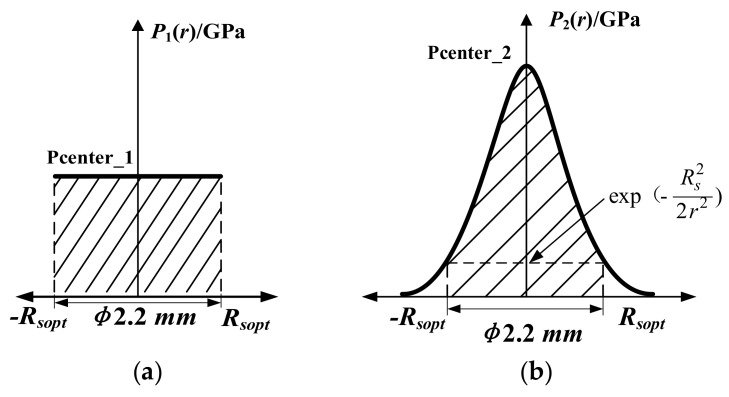
The spatial distribution of shock wave pressure: (**a**) flat-top spatial energy laser beam (FSELB); (**b**) Gaussian spatial energy laser beam (GSELB).

**Figure 5 materials-12-01343-f005:**
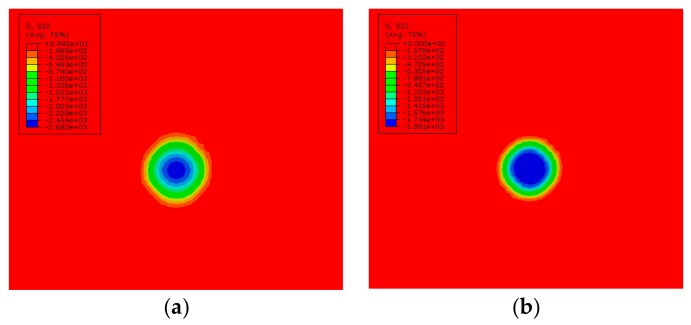
The residual stress of a single point: (**a**) GSELB; (**b**) FSELB.

**Figure 6 materials-12-01343-f006:**
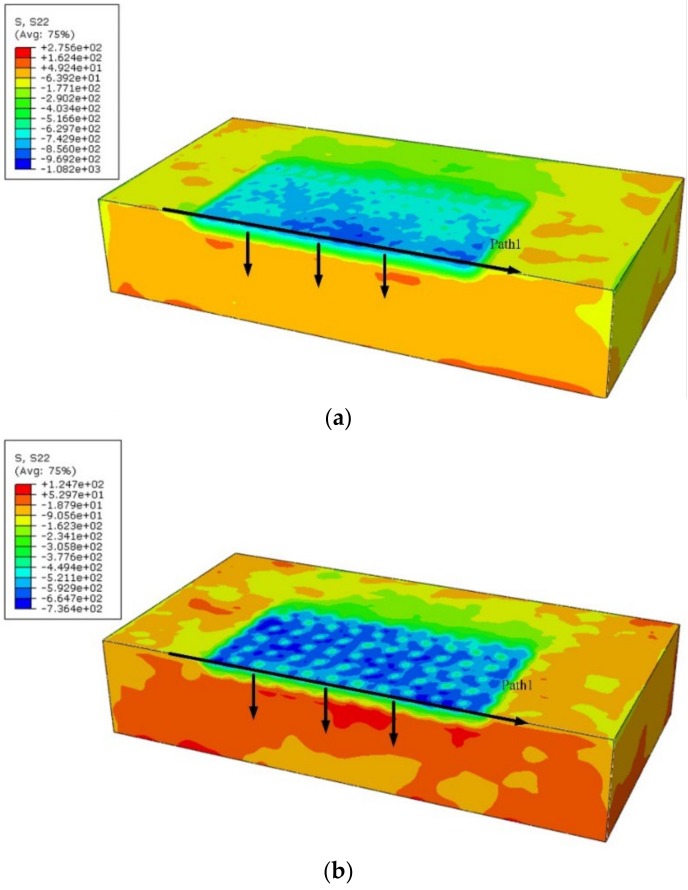
The residual stress field distribution of the finite element model (FEM) simulation: (**a**) GSELB; (**b**) FSELB.

**Figure 7 materials-12-01343-f007:**
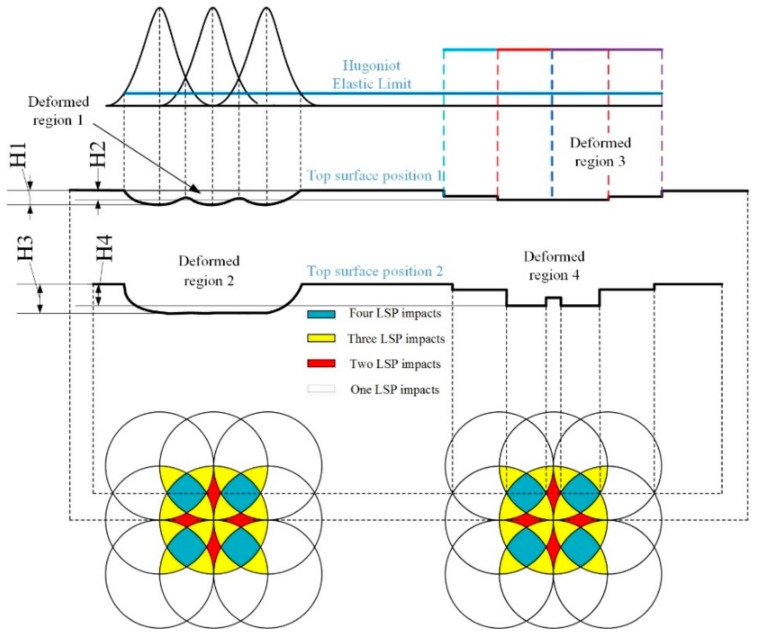
Schematic diagram of theoretical plastic deformation of titanium alloy after LSP.

**Figure 8 materials-12-01343-f008:**
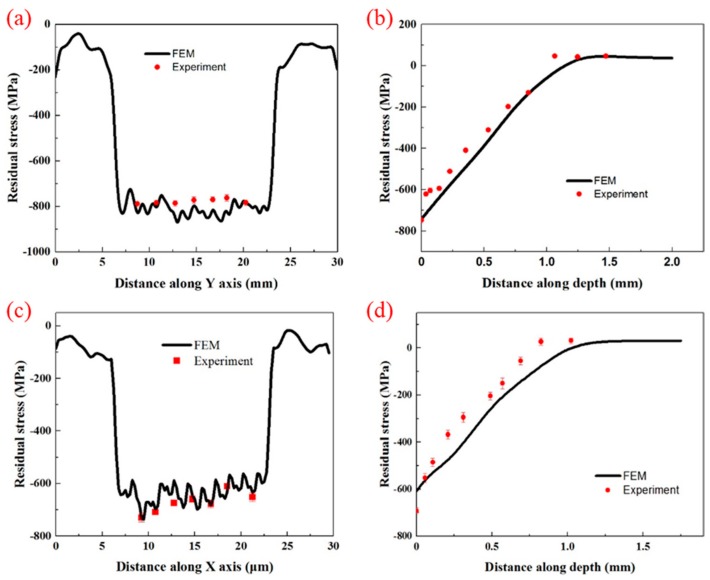
The residual stress field distribution of TC4 titanium alloy treated by LSP. (**a**) and (**b**) are the surface and depth residual stress treated by GSELB; (**c**) and (**d**) are the surface and depth residual stress treated by FSELB.

**Figure 9 materials-12-01343-f009:**
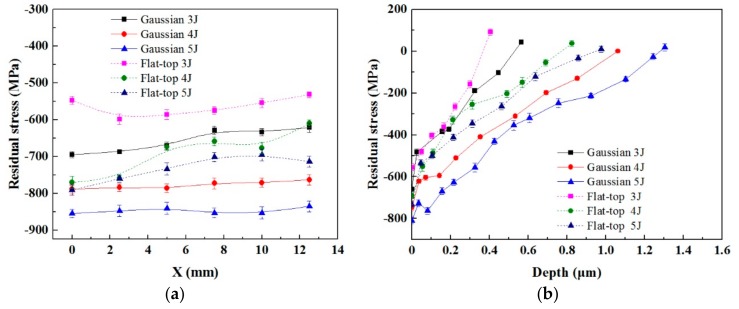
The distribution of residual stress: (**a**) surface; (**b**) depth.

**Table 1 materials-12-01343-t001:** Chemical composition of TC4 titanium alloy (%).

Composition	Al	V	Fe	C	N	H	O	Ti
Percent (%)	5.5–6.8	3.5–4.5	1.6–2.4	1.6–2.4	0.05	0.0125	0.13	Bal

**Table 2 materials-12-01343-t002:** Mechanical properties of TC4 titanium alloy.

Mechanical Property	Value
Density (g/cm^3^)	4.44
Poisson’s ratio	0.34
Elastic modulus (GPa)	109
Hugoniot elastic limit (GPa)	2.8

**Table 3 materials-12-01343-t003:** The detailed LSP parameters.

Parameters	Value
Laser wavelength (nm)	1064
Pulse energy (J)	3/4/5
Pulse duration (ns)	20
Spots diameter (mm)	2.2
Laser impact	1
Repetition-rate (Hz)	1
System ASE energy (mJ)	<50
Export laser energy stability	<±5%
Overlapping rate	50%

**Table 4 materials-12-01343-t004:** Johnson–Cook (J–C) model parameters of TC4 titanium alloy [29].

A (MPa)	B (MPa)	*C*	*n*	$\dot{\varepsilon }$ _0_	*m*
1098	1092	1.1	0.93	1	1.1

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
