# Peer review of "Simulation and Experimental Study on Residual Stress Distribution in Titanium Alloy Treated by Laser Shock Peening with Flat-Top and Gaussian Laser Beams"

_materials, 2019, doi:10.3390/ma12081343_

Reviewer 1 Report

Please find the attached report

Author Response

This manuscript presents a 3D finite element model to predict the residual stress with different spatial energy distribution and its validations using the experimental measurement, in addition to investigate the effect of LSP pulse intensity on the residual stress formed by flat-top and Gaussian laser beams. This title is highly recommended and required for publication as it will be important to different applications requirement.

The presented work has an interesting result and combines both modeling and experimental validation. Materials and experiments section are well presented. However, some points need to be addressed before being ready for publication as follows:

Q1: Shot peening is also one of the commonly used techniques to apply compressive residual stress. A critical review needs to be added to the introduction section, in order to focus on the importance of using LSP technique, novelty, and contribution of the current work as compared to literature studies. Some references are recommended such as follows:

- Liao Y, Suslov S, Ye C, Cheng GJ. The mechanisms of thermal engineered laser shock peening for enhanced fatigue performance. Acta Materialia. 2012 Aug 1;60(13-14):4997-5009.

- Maamoun, A.H.; Elbestawi, M.A.; Veldhuis, S.C. Influence of Shot Peening on AlSi10Mg Parts Fabricated by Additive Manufacturing. J. Manuf. Mater. Process. 2018, 2, 40.

- Hackel L, Rankin JR, Rubenchik A, King WE, Matthews M. Laser peening: A tool for additive manufacturing post-processing. Additive Manufacturing. 2018, 24: 67-75.

Response: Thank you for this comment. We added these references in the revised manuscript, which is to focus on the importance of using LSP technique.

Q2: The displaying order of Figures 6 and 8 regarding both GSELB and FSELB need to be adjusted to keep the displaying homogeneity as displayed in Figures 5 and 7.

Response: Thank you for this comment. We deleted the Figures 6 and 8 in the revised manuscript, which does NOT influence the conclusion of this paper.

Q3: For the results and discussion section, the claimed analysis does not have any support or reference using literature studies. This section needs to be comprehensively revised to include more discussion supported by previously published results in this field. Recent publications are highly recommended.

Response: Thank you for this comment. We added some recent publications in the revised manuscript.

Q4: The paragraph starts line #183 needs to be revised to clearly describe the results illustrated in Figure 9. Regions 2, 3, 4, 5 should be well defined in the manuscript as well.

Response: Thank you for this comment. We added the definition of regions 1, 2, 3, 4 in the revised manuscript, and rewrote the corresponding expression in this paragraph.

Q5: Figures 10 and 11 can be combined in only one figure to make it easy for the reader to compare the effect difference of both processes.

Response: Thank you for this comment. We combined the two figures in the revised manuscript.

Q6: Regarding the validation of the FEM and the obtained trend, the simulation results should be supported by minimum of two independent techniques. More measurement results should be added such as hardness measurements or microstructure characterization to investigate the validity of the simulation model.

Response: Thank you for the suggestion. The hardness measurement and microstructure characterization are important for the study of LSP, and we will further expand the research in the next. In this work, we focused on the difference on the residual stress distribution due to the different laser energy distribution, the measurement of hardness and microstructure does Not influence the conclusion of this paper.

Reviewer 2 Report

The authors present an interesting study on the laser shock peening of TC4 titanium alloy with two different laser beam energy distributions. First, they show how to simulate the LSP process by means of a FEM model. Then, they validate the simulation findings through experimentations. Moreover, they evaluate the dependence of the induced compressive residual stress on the pulse energy.

In general, the research study is quite well presented and I recommend to accept it after minor revisions:
Line 51: the sentence is not clear. There are more than two types of distributions of laser beam energy density. Why did the authors choose Gaussian and Flat-top? The authors can add some new reference explaining such a choice.
Line 102: in order to determine the final result with the FEM model the authors perform a springback analysis. It is not clear when it is done. What they mean with "this state"?

Line 109: The authors say that a coating material is used for the simulation. Is it used also in the experimental part? It can be added as information in the section 2.2 with a schematic representation as support in order to better clarify the setup.

Line 110: Do the authors refer to the Johnson-Cook model? Please specifiy what the capital letters mean.

Line 128: What is the laser pulse width? Are there any difference between gaussian and flat-top distributions? Please, clarify.

Line 172: the authors state that they use high power pulsed laser. This statemen is vague. What is the power? Different laser powers can provide different results.

Line 199: the authors mention a path in fig. 5. The current figure does not show any path and any black arrows. Please, clarify.

Line 204: the authors comment the results of the FEM model compared with the experiments. The reviewer suggests to add some information about the error between these results and about why the FEM model simulate a general trend of the residual stress lower than for the experiments.

Some english language revisions:
Line 36: extend to
Line 38: research studies

Line 46: without "the"

Line 69: Elastic modulus

Line 88: measured

Lines 108,112: strain

Line 142: without "width of"; t is the time

Line 174: largest

Line 251: treatment of.

Author Response

The authors present an interesting study on the laser shock peening of TC4 titanium alloy with two different laser beam energy distributions. First, they show how to simulate the LSP process by means of a FEM model. Then, they validate the simulation findings through experimentations. Moreover, they evaluate the dependence of the induced compressive residual stress on the pulse energy. In general, the research study is quite well presented and I recommend to accept it after minor revisions:

Q1: Line 51: the sentence is not clear. There are more than two types of distributions of laser beam energy density. Why did the authors choose Gaussian and Flat-top? The authors can add some new reference explaining such a choice.

Response: Thank you for this comment. We modified this sentence in the revised manuscript and made it more clearly, and we added three references about the Gaussian and Flat-top, which is to explain the reason we choose the two laser energy distribution.

Q2: Line 102: In order to determine the final result with the FEM model the authors perform a springback analysis. It is not clear when it is done. What they mean with "this state"?

Response: Thank you for this comment. We rewrote this sentence and made it more clearly to understand in the revised manuscript. The modified sentence is shown as follow:

ABAQUS/Explicit was first utilized to conduct dynamic calculation, which was implemented to capture the material response. After the dynamic calculation completion, the obtained results were input into ABAQUS/ Standard to perform static analysis, and then to output the final result.

Q3: Line 109: The authors say that a coating material is used for the simulation. Is it used also in the experimental part? It can be added as information in the section 2.2 with a schematic representation as support, in order to better clarify the setup.

Response: Thank you for this comment. We added the description of LSP experiment in the section 2.2, the corresponding is shown as follow:

In the LSP process, a water layer with about 1 mm thickness was used as the transparent confining layer and an Al foil with a thickness of 100 μm was used as the absorbing layer. The detailed principle of LSP is described by Ye et al. [5].

Q4: Line 110: Do the authors refer to the Johnson-Cook model? Please specifies what the capital letters mean.

Response: Thank you for this comment. We modified the expression in the revised manuscript.

Q5: Line 128: What is the laser pulse width? Are there any difference between gaussian and flat-top distributions? Please, clarify.

Response: Thank you for this comment. In Table 3 we set the pulse width is the same with 20 ns for the gaussian and flat-top distributions, and we modified the expression in the Line 128 in the revised manuscript.

Q6: Line 172: the authors state that they use high power pulsed laser. This statemen is vague. What is the power? Different laser powers can provide different results.

Response: Thank you for this comment. We deleted this sentence in the revised manuscript, , which does NOT influence the conclusion of this paper.

Q7: Line 199: the authors mention a path in Fig. 5. The current figure does not show any path and any black arrows. Please, clarify.

Response: Thank you for pointing this mistake. We modified the Fig. 5 into Fig. 6 in the revised manuscript.

Q8: Line 204: the authors comment the results of the FEM model compared with the experiments. The reviewer suggests to add some information about the error between these results and about why the FEM model simulate a general trend of the residual stress lower than for the experiments.

Response: Thank you for pointing this. We modified the expression and explained the reason of the difference between the FEM results and the experimental results.

Q9: Some english language revisions:

Line 36: extend to

Line 38: research studies

Line 46: without "the"

Line 69: Elastic modulus

Line 88: measured

Lines 108,112: strain

Line 142: without "width of"; t is the time

Line 174: largest

Line 251: treatment of.

Response: Thank you for pointing these mistakes. We corrected the typos and grammar mistakes in the revised manuscript.

Round  2

Reviewer 1 Report

The authors clearly addressed the review comments including a significant improvement of the revised manuscript.